# Modulating Microglia/Macrophage Activation by CDNF Promotes Transplantation of Fetal Ventral Mesencephalic Graft Survival and Function in a Hemiparkinsonian Rat Model

**DOI:** 10.3390/biomedicines10061446

**Published:** 2022-06-19

**Authors:** Kuan-Yin Tseng, Jui-Sheng Wu, Yuan-Hao Chen, Mikko Airavaara, Cheng-Yi Cheng, Kuo-Hsing Ma

**Affiliations:** 1Department of Neurological Surgery, Tri-Service General Hospital, National Defense Medical Center, Taipei 114, Taiwan; neuronsurgery@gmail.com (K.-Y.T.); chenyh178@gmail.com (Y.-H.C.); 2Department of Biology and Anatomy, National Defense Medical Center, Taipei 114, Taiwan; mosbyable@gmail.com; 3Drug Research Program, Faculty of Pharmacy, University of Helsinki, Viikinkaari 5E, 00014 Helsinki, Finland; mikko.airavaara@helsinki.fi; 4Neuroscience Center, HiLIFE, University of Helsinki, 00014 Helsinki, Finland; 5Department of Nuclear Medicine, Tri-Service General Hospital, Taipei 114, Taiwan

**Keywords:** cerebral dopamine neurotrophic factor (CDNF), ventral mesencephalic tissue, transplantation, positron emission tomography, Parkinson’s disease, microglia activation

## Abstract

Parkinson’s disease (PD) is characterized by the loss of dopaminergic neurons in substantia nigra pars compacta, which leads to the motor control deficits. Recently, cell transplantation is a cutting-edge technique for the therapy of PD. Nevertheless, one key bottleneck to realizing such potential is allogenic immune reaction of tissue grafts by recipients. Cerebral dopamine neurotrophic factor (CDNF) was shown to possess immune-modulatory properties that benefit neurodegenerative diseases. We hypothesized that co-administration of CDNF with fetal ventral mesencephalic (VM) tissue can improve the success of VM replacement therapies by attenuating immune responses. Hemiparkinsonian rats were generated by injecting 6-hydroxydopamine (6-OHDA) into the right medial forebrain bundle of Sprague Dawley (SD) rats. The rats were then intrastriatally transplanted with VM tissue from rats, with/without CDNF administration. Recovery of dopaminergic function and survival of the grafts were evaluated using the apomorphine-induced rotation test and small-animal positron emission tomography (PET) coupled with [^18^F] DOPA or [^18^F] FE-PE2I, respectively. In addition, transplantation-related inflammatory response was determined by uptake of [^18^F] FEPPA in the grafted side of striatum. Immunohistochemistry (IHC) examination was used to determine the survival of the grated dopaminergic neurons in the striatum and to investigate immune-modulatory effects of CDNF. The modulation of inflammatory responses caused by CDNF might involve enhancing M2 subset polarization and increasing fractal dimensions of 6-OHDA-treated BV2 microglial cell line. Analysis of CDNF-induced changes to gene expressions of 6-OHDA-stimulated BV2 cells implies that these alternations of the biomarkers and microglial morphology are implicated in the upregulation of protein kinase B signaling as well as regulation of catalytic, transferase, and protein serine/threonine kinase activity. The effects of CDNF on 6-OHDA-induced alternation of the canonical pathway in BV2 microglial cells is highly associated with PI3K-mediated phagosome formation. Our results are the first to show that CDNF administration enhances the survival of the grafted dopaminergic neurons and improves functional recovery in PD animal model. Modulation of the polarization, morphological characteristics, and transcriptional profiles of 6-OHDA-stimualted microglia by CDNF may possess these properties in transplantation-based regenerative therapies.

## 1. Introduction

Parkinson’s disease (PD), resulting from progressive loss of dopaminergic (DA) neurons in substantia nigra pars compacta, manifests resting tremor, muscular rigidity, bradykinesia, and postural instability [1,2,3]. Although the mainstream therapies for PD, including L-DOPA replacement or deep brain stimulation, could halt the neurodegenerative progression, they do not rescue the impaired dopaminergic system. In addition, long-term exposure of L-DOPA would cause debilitating involuntary choreic and dystonic movements when the therapeutic effectiveness of L-DOPA declines with time [4,5,6,7]. Therefore, cell replacement therapy is regarded as a potential treatment that complements the mainstream therapies for PD and aims to replace the lost DA functionally using exogenous cell sources [8,9,10,11]. Intrastriatal transplantation of human fetal ventral mesencephalic (hVM) tissue has been demonstrated to restore striatal dopamine level in PD patients and lead to the long-lasting symptomatic relief [12,13,14]. Nevertheless, host immune response toward the graft cells is still a major restriction that critically affects the success of transplantation [15,16,17,18]. Furthermore, poor integration efficiency of transplanted cells in the host dopaminergic system limits clinical applications [19,20]. Recently, the pathophysiology of PD has been implicated in microglial activation and proinflammatory microenvironments, which is considered to adversely affect the maturation and integration of VM grafts [21,22].

Microglia, peripheral monocytes/macrophages, and other innate immune cell types can both induce and resolve inflammation [23,24]. In the central nervous system (CNS), resident (microglia) and infiltrating innate immune cells coordinate a complex response toward the tissue repair. Proinflammatory type I microglia/macrophage (M1) can cause tissue damage, while anti-inflammatory or pro-reparative type II microglia/macrophage (M2) is implicated in resolution of inflammation and restoring tissue integrity [25,26]. Although dichotomy between M1 and M2 phenotypes has been used to describe the outcomes of immune cells in various brain diseases [27], canonical markers of both polarization states were highly co-expressed in the same cell under disease conditions, suggesting the possibility that different microglia subtypes, such as disease-associated microglia (DAM), may coexist [28]. Therefore, proper manipulation of the dynamic equilibrium among proinflammatory, disease-associated, and pro-reparative microglia/macrophages may be effective way to ameliorate host immune responses toward the graft cells, which critically affects the success of transplantation [16,29,30].

Cerebral dopamine neurotrophic factor (CDNF) is an evolutionarily-conserved protein with protective effects on midbrain dopaminergic system and cortical neurons in the ischemia model [31,32,33]. In addition, increasing evidence indicates that CDNF can protect the microglia against inflammatory injuries and alleviate the productions of proinflammatory cytokines, suggesting that CDNF might modulate microglial phenotypes to regulate inflammatory pathways in CNS disorders [34,35,36]. These data demonstrated that CDNF potentiates both neuroprotective and immunomodulatory properties in a number of small and large animal models of PD. However, there is still no report determining whether co-graft of CDNF would enhance the survival of the rat VM tissue or mitigate immune responses toward the graft cells in the striatum of the PD rats.

The current study is aiming to examine whether CDNF could enhance the survival of VM allografts in a PD model rat, and, if so, to reveal the underlying mechanisms of this improvement. In order to determine DA neuron viability, we used small-animal positron emission tomography (PET) with [^18^F]-labeled radioligands, [^18^F]-DOPA and [^18^F]-FE-PE2I, to quantify dopamine synthesis and the dopamine transporter, respectively. In addition, we used PET with [^18^F]-FEPPA, as a biomarker of neuroinflammation, to determine the temporal course of graft-induced inflammatory responses. We observed the microglia/macrophage activation and DA cell survival of different transplantation groups via PET imaging and immunohistochemistry studies. Furthermore, we used an apomorphine-induced rotation test to evaluate recovery of dopamine neuronal function. To shed light on the CDNF’s effect on microglial behavior, we turned to an approach of in vitro study of CDNF treatment in 6-OHDA-induced polarization of microglial phenotypes and subtleties of microglial morphology. To gain a comprehensive understanding of mechanisms to directly compare how CDNF treatment alters the transcriptome profile of BV2 cell lines following stimulation of 6-OHDA, RNAseq experiments were performed, and demonstrated that these alternations of microglial phenotype and morphology are implicated in the upregulation of protein kinase B signaling, catalytic, transferase, and protein serine/threonine kinase activity. Finally, we performed canonical pathways using QIAGEN’s Ingenuity^®^ Pathway Analysis tools and protein–protein interaction (PPI) network analysis. These unprecedented findings demonstrate that CDNF regulate the functional activation of microglia under pathological stress via upregulating PI3K/AKT signaling pathway, which is required for microglial phagosome formation. 

## 2. Materials and Methods

### 2.1. Animals

Male adult Sprague Dawley rats (280–300 g; BioLASCO, Taipei, Taiwan Co., Ltd.), based on previous experience involving similar experiment settings, were used for this study. Those animals were housed at the National Defense Medical Center’s Animal Center with a 12 h light/dark cycle, temperature of 25 ± 2 °C, 55% humidity, 2–4 animals per cage, and ad libitum standard diet and water. The experimental protocol was approved by the Institutional Animal Care and Use Committee (IACUC; protocol number 16258) of the National Defense Medical Center, Taiwan, R.O.C., which is accredited by the Association for Assessment and Accreditation of Laboratory Animal Care International (AAALAC International). All experiments were performed in a blinded manner, and the experimental results are reported according to the ARRIVE guidelines.

### 2.2. Hemiparkinsonian Rat Model

As previously stated, the hemiparkinsonian model rats were generated [37]. In brief, eight-week-old male SD rats were given 6-OHDA (20 μg in 4 μL of 0.02% ascorbic acid−holding saline) (Sigma-Aldrich, Saint Louis, MO, USA) unilaterally to the medial forebrain bundle (7.8 mm below the dura, 1.2 mm lateral to the midline, and 4.4 mm posterior to the bregma). The loss of approximately 90% of nigra DA neurons in this PD model rat has been demonstrated, together with severe apomorphine-induced rotational behaviors.

### 2.3. Behavioral Test

The apomorphine-induced rotational behavior test was used to assess the severity of damage in DA pathway in rats three weeks after 6-OHDA lesion [38]. Apomorphine was given subcutaneously to rats at a dose of 0.5 mg/kg in 0.2% ascorbic acid–holding saline (Sigma-Aldrich, Saint Louis, MO, USA). A rotometer system (MED Associates, Inc., St. Albans, VT, USA) was then used to record drug-induced rotational responses. Rats in a hemiparkinsonian model turned over four rotations per minute away from the lesion side. The rotation test was repeated at four and eight weeks after transplantation to assess DA functional recovery. 

### 2.4. Production of Recombination CDNF Protein

Recombinant human CDNF (rhCDNF), produced in mammalian cells, was purchased from Icosagen (cat. nr. P-100-100, Tallinn, Estonia).

### 2.5. Mesencephalic Tissue Preparation and Transplantation

Allotransplantation models were established using VM tissues taken from embryonic day 14 SD rats [17]. The dissection areas were chosen in accordance with previous researches. A large number of DA cells in dissected tissues was maintained in 1X HBSS. Using a glass micropipette, VM tissue was cut into minute slices and then implanted into lesioned striatum (0.5 mm posterior to bregma, 2.5 mm lateral to the midline, and 5.5 mm below the dura). Twenty-four hemiparkinsonian rats were separated into four groups in which grafted tissue or HBSS was delivered into the striatum, indicated in Figure 1A. (1) The vehicle group (*n* = 4) received 4 μL 1X HBSS followed by a weekly intracerebroventricular (I.C.V.) injection of 2 μL 1X PBS for 4 weeks. (2) The rhCDNF group (*n* = 5) received 4 μL 1X HBSS injection followed by I.C.V. administration of rhCDNF (5μg/2μL) weekly for 4 weeks. (3) The rVM group (*n* = 9) was implanted with rVM and then received a weekly I.C.V. injection of 2μL 1X PBS weekly for 4 weeks. (4) The rVM + rhCDNF group (*n* = 10) was implanted with rVM and subsequently given a weekly I.C.V. rhCDNF (5 μg/2 μL) injection for 4 weeks.

### 2.6. Radiopharmaceuticals

The Department of Nuclear Medicine native to National Taiwan University Hospital synthesized and provided [^18^F] DOPA [37]. [^18^F] FE-PE2I was synthesized with minor modification, as previously described. A thorough description of the synthesis of [^18^F] FEPPA has been described previously [39].

### 2.7. Small-Animal PET Imaging

The PET imaging procedure for small animals was based on previous research [37]. In brief, PET imaging was performed on rats using PET scanner (BIOPET 105, BIOSCAN, Santa Clara, CA, USA) at one week before the 6-OHDA lesion was produced, three weeks after the 6-OHDA lesion was created; four- and eight-weeks following transplantation. The rats were administered with [^18^F] DOPA (22.2–25.9 MBq; 0.6–0.7 mCi), [^18^F] FE-PE2I (14.8–18.5 MBq; 0.4–0.5 mCi), or [^18^F] FEPPA (0.9–1.1 mCi) through the tail vein. Entacapone (Toronto Research Chemicals, Toronto, ON, Canada; 10 mg/kg) and carbidopa (Toronto Research Chemicals, Toronto, ON, Canada; 10 mg/kg) were administered intraperitoneally 30 min before [^18^F] DOPA. PET image acquisition was performed after injection of [^18^F] FEPPA, 50 min and 20 min after radioligand injection of ([^18^F] DOPA) and ([^18^F] FE-PE2I), respectively. The data were collected in a range of energies between 250 and 700 keV. A 2D filtered back-projection (ramp filter, with the cutoff determined at Nyquist frequency) and the Fourier re-binning algorithm were used to reconstruct the image. PET scans were analyzed using Amide software (Stanford University, Santa Clara, CA, USA). Brain area confirmation was performed using the rat brain atlas and magnetic resonance imaging (MRI) templates. The atlas and MRI were also utilized to identify volumes of interest of the cerebellum and striatum, as seen in the reconstructed and summarized PET images. The specific uptake ratio (SUR) was calculated as (striatum–cerebellum)/cerebellum to assess [^18^F] DOPA or [^18^F] FE-PE2I uptake. Meanwhile, the standard uptake value (SUV), determining uptake of [^18^F] FEPPA, was expressed as regions of interest (ROI) X body weight/dose of radioligand injection (mCi).

### 2.8. Histology, Immunohistochemistry, and Image Acquisition

Rat brains isolated from four or eight weeks after transplantation were fixed in 4% paraformaldehyde (PFA) at 4 °C for at least one day. The isolated brain tissues were then immersed for two days in 20% sucrose in 0.1 M PBS followed by two days in 30% sucrose. Using a Cryostat Microtome (Leica CM 3050; Leica Microsystem, Wetzlar, Germany), brains were cut into a series of coronal sections (30 μm). All brain slices containing grafted region (approximately 90 slices per animal) were separated into four sets. The protocol of immunohistochemistry was described previously. Briefly, brain slices were washed with blocking solution comprising 3% normal goat serum (Vector, Burlingame, CA, USA) in PBS, and 0.5% Triton X-100 (Sigma-Aldrich, Saint Louis, MO, USA). Then, rabbit anti-TH (1:2000, Millipore), rabbit anti-DAT (1:500, Abcam), rabbit anti-Iba1 (1:500; Wako), and mouse anti-iNOS (1:250; Abcam) were chosen as first antibodies. Secondary antibodies conjugated with Alexa Fluor^®^ 488 or 568 (1:500; Invitrogen) were employed for fluorescence microscopy viewing. For nuclear quantification, the brain slices were stained with DAPI (1:1000; AAT Bioquest, Inc., Sunnyvale, CA, USA). Digital imagining was carried out on confocal microscopy (LSM880; Zeiss, Oberkochen, Germany) by means of a 10× and 40× objective. Microglia was counted as previously described [40]. The number of microglia was estimated for each rat by counting three successive brain sections including graft areas. The grafted regions were 850.2 μm^2^ in size. The density of activated microglia was calculated by dividing the total number of Iba1- and iNOS-immunoreactive cells by 850.2 μm^2^.

A biotinylated secondary antibody and peroxidase-conjugated streptavidin Vectastain ABC-detection system (Vector Laboratories) were employed to detect a signal by light microscopy. A slide scanner (Axio Scan.Z1; ZEISS, Oberkochen, Germany) and ImageJ software (version 1.8.0, NIH, Bethesda, MD, USA, 2014) were used to evaluate images of striatum stained with various antibodies. The densities of dopaminergic cells in grafted patches were estimated by dividing the total number of TH and DAT immunoreactive cells by the areas. Five coronal brain sections per animal (stereotaxic coordinates AP +1.6 to −0.8 mm; ML 0 to +5 mm; DV −0.2 to −8 mm) were imaged in the right striatum at two months after transplantation. The imaged regions were in the rVM graft-implanted dorsal striatum. As a result, cell number data were represented in terms of cells per cubic millimeter.

### 2.9. Cell Culture and Treatment

Murine BV-2 microglial cells were provided from Dr. Mei-Jen Wang [41] and cultured at 37 °C in Dulbecco’s Modified Eagle’s Medium (DMEM) containing 5% fetal bovine serum (FBS), 2 mM L-glutamine, 100 µg/mL streptomycin, and 100 U/mL penicillin (all from Gibco; Thermo Fisher Scientific, Inc., Waltham, MA, USA) under humidified 95% O_2_ and 5% CO_2_. The cells were then seeded into 24-well plates at a density of 1 × 10^5^/well and cultivated at 37 °C under humidified 95% O_2_ and 5% CO_2_. To determine the microglial morphology, the polarization, and gene expression, the medium was replaced with freshly prepared serum-free medium with or without nontoxic dose of 6-OHDA (30 µM) (Appendix A) and/or a dose of rhCDNF (1 μg/mL) for 24 h [42,43].

### 2.10. Immunofluorescence Staining and Analysis of Imaging Data

BV2 microglial cells were grown on the coverslips fixed with 4% PFA for 15 min at RT and washed three times with PBS. Cells were blocked in blocking buffer (5% bovine serum albumin (BSA), 0.3% Triton X-100 in PBS) for 1 h at RT. Cells were then incubated with the following primary antibodies overnight at 4 °C: anti-CD11b (monoclonal mouse; 1:250, Abcam), anti-Arg1 (polyclonal rabbit; 1:200; Santa Cruz) and then secondary antibodies conjugated with Alexa Fluor^®^ 488 or 568, (1:500, Life Technologies, California, United States) were used. Finally, coverslips were washed three times in PBS, stained with 4′-6-diamidino-2-phenlyindole (DAPI, Sigma-Aldrich) 5 µg/mL in PBS, and mounted in fluorescent mounting medium (SHANDON, ThermoFisher Scientific Inc., Waltham, MA, USA). Fluorescence images were captured with Zeiss AxioImager M2 482 epifluorescence microscope equipped with a 483 AxioCam HRm camera. Images were acquired with the AxioVision4 software. For fluorescent intensity measuring, 8-bit grayscale images were taken at 10× magnification. Mean fluorescence values were measured in 6-OHDA-treated BV2 cells with or without rhCDNF administration, and after background subtraction, the values for 6-OHDA-treated BV2 cells were normalized to those of BV2 cells with PBS treatment. In vitro experiments were performed and analyzed 6–7 times.

### 2.11. Western Blot Analysis

Cells were homogenized in NP-40 lysis buffer (50 mM Tris-HCl, pH 7.4, 0.15 M NaCl, 1.0 mM EDTA, 1% NP-40) freshly supplemented with Phosphatase and Protease Inhibitor Cocktail (Complete, Mini, EDTA-free, Roche Life Science), incubated on ice for 15–30 min and centrifuged at 13,000 rpm for 10 min. Protein aliquots were electrophoresed on sodium dodecyl sulfate–polyacrylamide gels (8% or 12%) and transferred to a polyvinylidene difluoride membrane. The membranes were rinsed in 0.01 M Tris-buffered saline (pH 7.4) containing 0.1% Triton X-100 for 10 min, blocked in 5% non-fat dry milk for 30 min, and then incubated with IL-10 (rabbit polyclonal; 1:500; Santa Cruz), Arginase-1 (rabbit polyclonal; 1:500; Santa Cruz), iNOS (rabbit polyclonal; 1:500; Abcam), CD11b (mouse monoclonal; 1:250; Abcam), IL-6 (rabbit polyclonal; 1:250; Abcam), or GAPDH (1:1000, Abcam) overnight at 4 °C. Following three washes with Tris-buffered saline, the membrane was immersed in Tris-buffered saline containing 5% non-fat dry milk containing a horseradish peroxidase-conjugated secondary antibody for 2 h at room temperature. Enhanced chemiluminescent autoradiography (ECL kit; Amersham Life Science, Arlington Heights, IL, USA) was used for acquisition of immunoreactivity. The Western blots were photographed using a digital camera, and the intensities were calculated using NIH Image J.

### 2.12. Fractal Analysis Using FracLac for ImageJ

For determining the morphological changes of microglia at one month after transplantation, photomicrographs were acquired from the grafted striatum, where microglia activation was expected to be high. As previous studies showed, fractal analysis (FracLac for Image J) was adopted to quantify the complexity of Iba1-immunoreactive microglia [44]. In brief, a total of 30 cells was chosen for fractal analysis per animal in each photomicrograph (6 photomicrographs per animal). For in vitro study, we also utilized fractal analysis to investigate the morphological alternations of CD11b-immunoreactive BV2 microglia following 6-OHDA or rhCDNF treatment [40,44]. Within each photomicrograph, five microglia were chosen at random for this analysis (4 photomicrographs per well). A total of 100 cells was selected for fractal analysis (using a grid and random number generator) per group. ImageJ was utilized to convert binary images to depict outlines. Fractal dimension (fD) is the measure of microglia complexity that quantifies the contour of each cell defined by the endpoints and process lengths. FracLac for ImageJ determines the microglia fD for each cell, as previously reported [44,45]. 

### 2.13. Assessment of Cytokines

Tissues from a total of 10 rats were used for the cytokine assay by enzyme-linked immunosorbent assay (ELISA). The grafted striatum was collected at one month after transplantation. After homogenization in lysis buffer (PRO-PREP™, iNtRON Biotechnology, Korea) and centrifugation at 15,000 g for 30 min, we collected the supernatants and stored them at −80 °C. During quantification, the cytokines (TNFα, IL-1β, IL-6) were normalized to 100 μg of protein in the supernatant using a commercial ELISA kit (R & D Systems, Minneapolis, MN, USA) according to the manufacturer’s instructions.

### 2.14. RNA Preparation and RNA Sequencing

Population RNA-seq was carried out as previously described [40]. Briefly, 10^5^–10^6^ BV2 cells from each group were harvested and kept at 80 °C in 50 µL of lysis/binding buffer (Life Technologies). RNA was isolated using Trizol reagent and processed with DNase (#1906, Ambion, ThermoFisher Scientific Inc., Waltham, MA, USA) and further purified with the Qiagen RNeasy MiniElute Cleanup kit (Qiagen, #74204). The NEBNext Ultra 435 Directional RNA Library prep kit (NEB, #E7420L), 14 cycles of PCR amplification, and single (i7) indexing were adopted for purified mRNA library preparation. Indexed library preps from each sample were assembled and 437 sequenced using a NextSeq High Output 75 cycle flow cell (Illumina) with 75SE reads at a pool concentration of 1.3 pM. The 439 Illumina bcl2fastq (v2.20.0.422) software was conducted for base calling and demultiplexing. STAR aligner (2.6.0c) was used to map reads to zebrafish GRCz11 genome 440. The featureCounts tool from Subread 441 package (v1.22.2) was utilized to determine gene counts from Ensembl release 97 zebrafish gtf-files.

### 2.15. Bioinformatics

CASAVA base calling was used to convert the original data from high-throughput sequencing (Illumina NovaSeq 6000 platform) into raw sequenced reads, which were then stored in FASTQ format [46,47]. These results have been entered into the NCBI BioProject database (https://www.ncbi.nlm.nih.gov/sra accessed on 22 February 2022). The NCBI SRA accession number for these data is PRJNA821327. After removing low-quality reads and bases [48,49,50], the resultant high-quality data were used for subsequent analysis. Read pairs from each sample were coordinated to the reference genome using the HISAT2 software (v2.1.0) [49,50]. For gene expression, “trimmed mean of M-values” normalization (TMM) and “relative log expression” normalization (RLE) were conducted using DEGseq [46,47,51]. DEGseq (without biological replicate) and DESeq2 (with biological replicate) were used in R to analyze differentially expressed genes (DEGs) analysis of two conditions, which are based on the negative binomial distribution and Poisson distribution model, respectively [52,53,54]. The obtained *p*-values were modified using Benjamini and Hochberg’s approach for controlling the FDR. CLC Genomics Workbench (version 21) identified differentially expressed genes, and differences with an FDR *p*-value < 0.001 and fold change >2 or <−2 were considered as significant [55,56]. ClusterProfiler (v3.10.1) was used for GO enrichment analysis of DEGs [57]. FDR values less than 0.001 were considered to be greatly enriched in the annotation category. An Ingenuity Pathway (IPA^®^, QIAGEN Redwood City, www.qiagen.com/ingenuity accessed on 22 February 2022) was conducted to analyze the most significant canonical pathways in the datasets as preciously described [46,58]. For literary analysis, genes from datasets related to canonical pathways in the Ingenuity Pathways Knowledge Base (IPAKB) were detected. Following prior methodologies, the significance of the connections between datasets and canonical pathways was examined [59]. Gene identifiers were assigned to corresponding gene objects when the datasets were uploaded, and the genes were overlaid into a global molecular network in the IPAKB. Based on connectivity, gene networks were constructed algorithmically [60].

### 2.16. Statistical Analysis

All graphs and statistics were performed in GraphPad Prism 9. PET images SURs, behavioral test results, and cell counts in the grafted areas derived through IHC staining were compared between groups. ANOVA with Bonferroni post-test was used for multiple comparisons, and Student’s *t*-test was used to compare the results of two independent groups. Values of *p* < 0.05 were considered to indicate statistical significance. Values are presented as mean ± S.E.M. A statistically significant difference was defined as *p* < 0.05.

## 3. Results

### 3.1. CDNF Boosts the Effect of rVM Transplantation on Functional Recovery in Hemiparkinsonian Rats

Apomorphine rotation was adopted to confirm hemiparkinsonian rats after the 6-OHDA lesion. There was no significant difference in initial apomorphine-induced rotation after 6-OHDA lesions between each group. However, less rotational behavior was obvious in the PD model rats receiving rVM and rVM + rhCDNF grafts at 1 or 2 months after transplantation, indicating that animals of these two groups have functional recovery compared with vehicle and rhCDNF groups (Figure 1B). To further evaluate grafted DA function after transplantation, the ratio of apomorphine-induced rotation number at 1 or 2 months after transplantation to that after 6-OHDA lesion was analyzed in each of the hemiparkinsonian rats. Although both rVM and rVM + rhCDNF grafts led to a significant reduction in the ratio of rotation number in the PD model rats, rhCDNF-treated rVM grafts appeared to produce a profound improvement in rotational asymmetry behavior at 2 months after transplantation, compared to rVM grafts alone (Figure 1C). Therefore, rats that received rVM + rhCDNF exhibited a continual motor improvement compared to rVM-grafted rats, suggesting that rhCDNF as a co-treatment material could enhance the effect of VM allotransplantation.

### 3.2. CDNF Improved the Functional Result of DA Allografts in the Striatum of Hemiparkinsonian Rats at 2 Months after Allotransplantation

The functional results of DA allotransplantation in hemiparkinsonian rats were evaluated by [^18^F] DOPA for PET scans. The images were acquired before and after establishment of the 6-OHDA lesion as well as at 1 and 2 months after transplantation. A 6-OHDA injection resulted in a decrease in the [^18^F] DOPA uptake level on the right side of striatum, as compared with that before 6-OHDA injection. The SUR of [^18^F] DOPA of the right striatum was quantified to further determine the effects of transplantation and 6-OHDA lesion. In our previous study, the SUR of [^18^F] DOPA of the 6-OHDA-lesioned striatum was shown to drop 85–95% from the baseline value. In the vehicle group, there was no recovery of the [^18^F] DOPA uptake in the period of 2-month follow-up after 6-OHDA injection (Figure 2A). In addition, there was no statistical improvement in [^18^F] DOPA uptake of the rhCDNF group at 2 months after HBSS injection, suggesting that single administration of rhCDNF after 6-OHDA injection does not promote regeneration of DA cells (Figure 2B). However, the uptake of [^18^F] DOPA significantly upregulated in the rVM tissue graft groups at 1 and 2 months after transplantation. Importantly, rhCDNF co-treated with rVM tissue elicited higher [^18^F] DOPA uptake than that of rVM graft group at 2 months after transplantation (Figure 2A,B). The above results imply that rhCDNF can further enhance the DA synthesis of allotransplantation of VM tissue grafts in hemiparkinsonian rats.

### 3.3. CDNF Promotes Maturation of DA Allografts in the Striatum of Hemiparkinsonian Rats at 2 Months after Transplantation 

[^18^F] FE-PE2I is a valuable radioligand to quantify dopamine transporter expression and determine the maturation of dopaminergic neurons from cell replacement [61]. In line with [^18^F] DOPA imaging results, uptake levels of striatal [^18^F] FE-PE2I on the lesioned area were decreased after 6-OHDA injections, which were recovered in the rVM or rhCDNF + rVM group. The SURs of [^18^F] FE-PE2I were reduced after 6-OHDA injections in all groups (Figure 3A). Administration of vehicle or rhCDNF in the lesioned striatum did not significantly increase the uptake of [^18^F] FE-PE2I, which is comparable to the [^18^F] DOPA imaging results. However, the SURs of [^18^F] FE-PE2I in the rVM and rVM + rhCDNF groups were recovered after transplantation. Notably, co-treatment of rhCDNF and rVM tissue further enhanced uptake of [^18^F] FE-PE2I at 2 months after transplantation (Figure 3A,B). This result suggests that rhCDNF co-treated with rVM tissue can promote maturation of transplanted DA cells in allotransplantation. 

### 3.4. CDNF Enhances the Survival and Maturation of DA Neurons in the rVM Grafted Striatum

To clarify whether the higher SURs of [^18^F] DOPA attribute to more DA cells in the grafted tissue, we observed TH immunostaining of the lesioned striatum at 2 months after transplantation. In photomicrographs of brain sections of the vehicle and rhCDNF groups, TH-stained cells or fibers in the lesioned striatum were scarcely visible (Figure 4A,B). In contrast, TH-stained cell bodies and fiber were detectable in the striatum of rVM or rVM + rhCDNF transplantation groups (Figure 4C,D), while TH-stained cells or fibers were barely found in the substantial nigra of the grafted side (Appendix A). Then, we quantified these IHC results by counting the number of TH-positive cell bodies in the lesioned striatum. With the vehicle group (22 ± 2/mm^3^) being the standard, we found that rVM (2584 ± 307/mm^3^) and rVM + rhCDNF (6919 ± 261/mm^3^) groups contained abundant TH-positive cell bodies over the grafted striatum. Accordingly, co-treatment of rhCDNF and rVM led to more TH-positive cells in the grafted striatum than that of rVM group (Figure 4E). This observation indicated that rhCDNF can provide long-term beneficial effects to the grafts. Next, DAT immunostaining of brain sections was performed to investigate whether CDNF further promotes the grafted DA cell maturation in the striatum. Although DAT-immunoreactive cells were observed in the grafted striatum of the rVM and rVM + rhCDNF groups, there was significantly higher DAT-immunoreactive cell body density in the grafted striatum of rVM + rhCDNF group (3478 ± 169/mm^3^) as compared with that of rVM group (1267 ± 171/mm^3^) (Figure 5A–C). These data not only corroborated previous results of the PET imaging, but indicated that rhCDNF, when co-treated with rVM tissues, can boost maturation of transplantation DA cells.

### 3.5. CDNF Attenuates Neuroinflamm. in Grafted Striatum after Allotransplantation 

The benefit of CDNF on grafted DA cell survival and maturation could be caused by direct neuroprotective activity of CDNF or an indirect effect of CDNF on the microenvironment of the lesioned striatum. To clarify underlying mechanisms, we tested if CDNF could modulate inflammatory responses after transplantation. [^18^F] FEPPA has been shown to offer a superior tool to quantify the expression of translocator protein (TSPO), which is located on outer mitochondria membranes in microglia and may represent a potential in vivo biomarker of neuroinflammation and reactive gliosis [39,62]. At 1 or 2 months after 6-OHDA injections, the uptake of [^18^F] FEPPA on the lesion side of striatum was observed, suggesting that 6-OHDA-induced DA denervation in the striatum induces long-term inflammatory responses. To further determine the graft-induced immune responses, the SUR of [^18^F] FEPPA of the lesioned side of the striatum was quantified. In the vehicle or rhCDNF groups, there was no difference in [^18^F] FEPPA uptake in the period of 2-month follow-up after HBSS injection (Figure 6A). However, the uptake of [^18^F] FEPPA of lesioned striatum significantly increased in the rVM tissue graft groups at 1 month, not at 2 months, after transplantation, implying that rVM induces greater neuroinflammation in the striatum at an earlier time point (Figure 6B). Notably, rhCDNF co-treated with rVM tissue led to less uptake of [^18^F] FEPPA in the lesioned striatum than that of rVM graft group at 1 month after transplantation (Figure 6A,B). This result suggests that rhCDNF can promote long-term DA survival and maturation in the grafted tissue, likely due to early mitigating graft-induced inflammatory responses.

To test whether CDNF’s immune modulatory function is related to influence microglial responses, we evaluated microglial phenotypes in grafted striatum via immunostaining brain slices with Iba1 (total microglia/macrophage), iNOS (activated innate immune cells), or double-stained with Iba1 and iNOS (classical pathway-activated microglia/macrophage). We counted the number of Iba1- and iNOS-positive cells per mm^2^ in the grafted striatum. In hemiparkinsonian rats that received rVM (*n* = 5), total microglia/macrophages in the grafted striatum were 263 ± 93/mm^2^ (Figure 7A) and activated innate immune cells were 81 ± 37/mm^2^ (Figure 7C), respectively. In rats that received rVM + rhCDNF (*n* = 5), total microglia/macrophages in the grafted striatum were 67 ± 20/mm^2^ (Figure 7B) and activated innate immune cells were 30 ± 5/mm^2^ (Figure 7D), respectively, suggesting that CDNF can attenuate recruitment of inflammatory and immune cells in the grafted striatum (Figure 7E,F). Importantly, rats that received rVM co-treated with rhCDNF had a lower density of classically activated microglia/macrophages (Iba1 and iNOS merged cells) than rats in the rVM group (Figure 7G–I). Next, we sought to characterize the morphological responses of microglia following the transplantation insult. Examples of Iba1-positive microglia (made binary and outlined) in the grafted striatum with/without rhCDNF administration are shown in Figure 8A–D. The application of FracLac for Image J to microglia outlines fractal dimension that ranged from 1.463 to 1.879, with the lowest occurring in the right striatum of rVM group at 1 month after transplantation and the highest in the left striatum with the corresponding groups. There was no difference in the fractal dimensions of Iba1-immunolabeled cells in the left striatum with or without rhCDNF administration (Figure 8A,C,E). However, one-way ANOVA analysis revealed that CDNF treatment could increase the fractal dimensions of Iba1-labeled microglia in the grafted striatum (right striatum) at one month after transplantation, compared with the rVM group (Figure 8B,D,E). These results indicated that CDNF mitigates grafted-induced inflammatory responses via not only decreasing microglia/macrophage recruitment, but also regulating its polarization with proinflammatory phenotype and morphological characteristics in allotransplantation. 

Since CDNF was shown to mitigate proinflammatory activation of microglia following transplantation, we further tested the efficacy of intracerebroventricular rhCDNF treatment on the cytokine responses in the grafted striatum. The administration of rhCDNF resulted in a significant decrease in IL-1β or TNF-α within the grafted striatum at one month after transplantation (Figure 8F,G). Meanwhile, the level of IL-6 within grafted striatum exhibited no statistically significant difference between rVM and rVM + rhCDNF groups (Figure 8H), implying that rhCDNF treatment predominantly suppressed microglial activation-induced proinflammatory cytokines in the rVM transplant. 

### 3.6. CDNF Modulates the Phenotype and Morphology of 6-OHDA-Stimulated BV2 Microglial Cell Lines

To test whether CDNF could directly regulate the polarization of microglia under stimuli, BV2 microglial cells were challenged with 30 mM 6-OHDA in the presence or absence of 1 μg/mL rhCDNF for 24 h. By quantifying the fluorescence staining intensity, 6-OHDA-treated BV2 cells stained higher, intensely, with CD11b (classically activated microglia marker), but less with Arg1, compared to control group (PBS-treated BV2 cells) (Figure 9A,B). This result suggests that 6-OHDA treatment led to BV2 microglial cell polarized to proinflammatory, rather than tissue-reparative, phenotype. However, administration of rhCDNF (1 μg/mL) increased the fluorescent intensity of Arg1 accompanied by decreasing CD11b expression in 6-OHDA-treated BV2 cells. By quantitative immunoblotting analysis, the levels of IL-10 in 6-OHDA-treated BV2 cells were decreased when compared to the PBS group. In contrast, 6-OHDA-treated BV2 cells exhibited higher proinflammatory markers (iNOS, CD11b, and IL-6) (Figure 9C). In line with immunostaining results, CDNF supplementation suppressed the expressions of CD11b, iNOS, and IL-6, but upregulated IL-10 and Arg-1 levels in 6-OHDA-treated BV2 cells (Figure 9D). These results suggest that CDNF protein supplementation would induce 6-OHDA-treated BV2 microglial cells polarizing into the alternative activated microglia. In addition to modulation of microglia phenotypes, we want to investigate if CDNF further regulates morphologic responses of BV2 cells under 6-OHDA stimuli. FracLac for Image J outlining BV2 cell morphology was applied to calculate fractal dimensions that ranged from 1.457 to 1.68 (available range is 1–2), with the lowest occurring in the 6-OHDA-treated group and the highest in the PBS-treated group. There was no difference in the fractal dimensions of Iba1-immunolabeled BV2 cells with or without rhCDNF administration (Figure 9E,F). However, one-way ANOVA analysis revealed that rhCDNF treatment increases the fractal dimensions of BV2 cells incubated with 6-OHDA, when compared to the solely 6-OHDA-treated cells (Figure 9G–I). The above results suggest that microglial complexity is unaffected with or without CDNF supplement, whereas cell complexity was lower after 6-OHDA incubation and was recovered approximately to the unstressed condition through CDNF therapy. Based on qualitative observations of the Iba1-labeled cell morphological characteristic, it was suggested that CDNF supplement can redirect microglial phenotype and morphology in the 6-OHDA-stimulated condition.

### 3.7. CDNF Treatment Alters Transcriptional Responses and Canonical Pathways in 6-OHDA-Stimulated BV2 Microglial Cells

Since rhCDNF treatment has effects on the phenotypical and morphological changes of microglia under stimuli, we naturally hypothesized that CDNF would alter the transcriptomic profile of 6-OHDA-treated BV2 cell lines. Gene profiles between PBS- and rhCDNF-treated groups were analyzed in the BV2 cell lines exposure to 24 h duration of 6-OHDA (Figure 10A). Next, principal component analysis (PCA) using DESeq2 showed a good separation among biological replicates of the population in PBS-treated and rhCDNF-treated BV2 cell lines with or without exposure to 6-OHDA. Volcano plots that graphically illustrate the DEGs that were significantly regulated (red) in response to rhCDNF treatment and PBS treatment in the BV2 cell lines exposure to 6-OHDA were generated (Figure 10B). According to the above criteria, 253 genes were differentially regulated in rhCDNF-treated BV2 cell lines exposure to 6-OHDA for 24 h. Of these, rhCDNF supplement upregulated 72 genes (Appendix A), but downregulated 181 genes (Appendix A) after 6-OHDA treatment in BV2 cell lines. In this analysis, we adopted triplicate RNA-seq and FDR value less than 0.001 for differential expression genes, which focuses on significantly differential expression genes in BV2 cell lines after 24 h 6-OHDA incubation. To exhibit the functional categories of DEGs, gene ontology (GO) enrichment analysis of DEGs (6-OHDA + vehicle vs. 6-OHDA + rhCDNF) identifies crucial biological processes. The significantly upregulated functional categories (FDR < 0.05) in the rhCDNF-treated group were involved in protein kinase B signaling, multicellular organismal processes, regulation of catalytic, transferase, and kinase activities; and immune system process (Figure 10C). Since the downregulated genes were not correlated to inflammation, only the upregulated genes were further studied. To further shed light on the molecular functions of regulated genes, IPA (IPA, Ingenuity Systems, http://www.ingenuity.com accessed on 22 February 2022) was used to identify the canonical pathways that characterize the relevant molecular functions based on functional knowledge inputs. Based on IPA analysis of transcriptome profiling data, the top five canonical pathways for differentially expressed genes between 6-OHDA + vehicle and 6-OHDA + rhCDNF groups are displayed in Figure 10D. The most significant canonical pathways were involved in phagosome formations and pattern recognition receptors in recognition bacteria and viruses. Other notable pathways included FAK and G-protein coupled receptor signaling in BV-2 cells co-treated with rhCDNF and 6-OHDA. The genes in the network (Figure 10E) were involved in phagosome formation at 24 h after rhCDNF treatment in 6-OHDA-stimulated BV2 cells. Interestingly, the genes enriched in networks were PIK3 and SRC complex, forming the central nodes of an interconnected regulatory system. Among these upregulated genes in volved in phagosome formation, PIK3CD, PIK3CG, TLR2, TLR3, and TLR8 have been implicated in the role of pattern recognition receptors in recognition of bacteria and viruses.

## 4. Discussion

In this study, we first utilized three radioligands coupled with animal PET images to characterize the two-month progression of DA function in rVM tissues and neuroinflammation in grafted striatum. We provide evidence that intracerebral delivery of rhCDNF following implantation of rVM resulted in better functional recovery of drug-induced rotational behavior and DA function after transplantation in hemiparkinsonian rats. Moreover, co-treatment of transplanting rVM tissues and rhCDNF not only increased dopamine synthesis but induced higher DAT expressions at 2 months after transplantation in the grafted striatum, indicating that delivery of rhCDNF can promote long-term survival of DA neurons and their maturation in rVM tissues. In line with results of PET images, rhCDNF treatment was shown to increase the numbers of TH-immunoreactive and DAT-immunoreactive neurons in the rVM-grafted striatum. However, the question arises as to why the effect of CDNF treatment on behavioral alternations was less distinct than promoting the function of DA neurons in the grafted striatum. One possible reason involves trophic effects of progenitor cells in rVM. Previous studies have reported that only VM into syngeneic recipients has the ability to improve apomorphine-induced rotational behavior in hemiparkinsonian rats and induce the recovery in the striatal uptakes of [^18^F] DOPA after transplantation [37,63], suggesting that progenitor cells in rVM tissues are able to synthesize dopamine in the lesioned striatum to alleviate 6-OHDA-induced behavior alternations. However, the striatal uptake of [^18^F] FE-PE2I was not recovered significantly in rVM-implanted striatum, implying that these cells in rVM tissues are still exhibited as DA precursor cells, not developed into DAT-expressed matured DA neurons [37]. While rhCDNF treatment did not further alleviate behavior alternations at an earlier time point, which might be reflective of DA precursor cell function, this therapeutic application could predominantly promote DA maturation representative of increase in [^18^F] FE-PE2I uptake in rVM-implanted striatum. This regenerative capacity of CDNF is comparable to the previous study demonstrating that mesencephalic astrocyte-derived neurotrophic factor, MANF, a paralogous protein of CDNF, has pro-reparative effects to enhance transplanted cell survival [64,65]. 

Abundant studies demonstrated that all grafts can induce activated immune cells migrating to the transplant side to cause inflammatory responses in host striatum, and more extensively in the xenograft conditions [16,66]. Although activated microglia/macrophages clear debris resulting from dead cells to improve local microenvironment, long-term activation of the innate immune system leads to graft destruction through antigen presentation and lymphocytic activation [67,68]. Previous studies showed that in graft conditions, microglia/macrophages act as antigen-presenting cells in the brain parenchyma, which was sufficient enough to interfere with graft survival and integration [36,69]. In addition, transplantation procedures, such as a posttraumatic brain injury, could involve the induction of proinflammatory and pro-reparative genes. Pro-reparative or immunosuppressive microglia/macrophages, promoting remyelination and repair after spinal cord injury and stroke, were transient and in most cases returned to preinjury level by 48 h postinjury [70,71]. In contrast, proinflammatory microglia/macrophages, producing proinflammatory mediators, reactive oxygen species, nitric oxide, proteolytic enzymes, and chemokines, were maintained for up to one month after injury [16,28,72,73]. The uncontrolled proinflammatory mediators lead to further grafted area damage and dysfunction. Therefore, the precise coordination of microglia/macrophage phenotype plays a crucial role in the survival and differentiation of progenitor cells for the efficient repair of damaged areas. In our work, a neuroinflammatory response, evaluated by uptake of [^18^F] FEEPA in small-animal PET, is increased in the grafted striatum at one month after rVM transplantation. However, this inflammatory response subsided to the basal level of vehicle group at two months after transplantation, suggesting that graft implantation can induce a strong, but short-lived, host immune response. In this context, the administration of rhCDNF significantly mitigated this graft-induced neuroinflammation. The anti-neuroinflammatory effect of CDNF was suggested to enhance DA neuron function and maturation in the grafted striatum at 2 months post-transplantation. Furthermore, this attenuation of inflammatory responses may involve the suppression of immune cell recruitment and modulation of the microglia/macrophage polarities in the grafted striatum. 

CDNF has been known as an ER lumen-resided protein with potent cytoprotective properties in numerous cell types. It can be secreted under cellular stress and can protect neighboring cells in paracrine fashion [32,74,75]. The mode of action of CDNF mainly alleviates ER stress-induced cell damage and activates the PI3K-Akt pathway, which is implicated in cell survival [76]. In addition to promoting neuronal survival in the toxin-based models of PD, transient expression of CDNF was shown to decrease production of IL-6 after 6-OHDA lesions [77]. In the SCI model, delivery of bone-marrow-derived mesenchymal stem cells (BMSCs) overexpressing CDNF promoted nerve regeneration and decreased levels of proinflammatory cytokines to attenuate neuroinflammation [78]. While these studies reveal that CDNF anti-inflammatory capacity is observed in vivo, they do not elucidate whether this effect merely attributes to less cell damage or a mechanism involved in phenotype changes of immune cells, such as microglia or recruited macrophages. In the current study, CDNF treatment was shown to predominantly decrease Iba1/iNOS double-positive microglia/macrophages recruiting into grafted striatum, implying that CDNF may induce immune privilege to possess neuroprotective effects on grafted cells. 

Although some studies suggest that CDNF can directly reduce lipopolysaccharide-induced secretion of PGE2 and IL-1β by inhibition of JNK pathways in cultured microglial cells [43], other in vitro studies argue that astrocyte-derived CDNF is not involved in preventing microglial activation [79]. In this study, we focused on the effects of 6-OHDA treatment on BV2 microglial cell lines and explored related mechanisms to clarify CDNF’s effects on microglial reaction in the neurotoxin-induced models of PD. Our findings demonstrate that proinflammatory mediator expressions, such as iNOS and IL6, in cultured BV2 microglia cells induced by neurotoxin 6-OHDA could be prevented by pretreatment with rhCDNF. In contrast, 6-OHDA abolished the expression of IL-10, which was recovered by treatment with rhCDNF in cultured BV2 microglia cells. In addition, CDNF supplementation could enhance Arg1 expression in 6-OHDA-siumulated BV2 cell lines, suggesting that CDNF can boost microglial polarization toward anti-inflammatory phenotype serving as a reparative role in neurodegenerative diseases [42]. Generally, the pro-reparative activation state comprises consecutive responses related to healing and scavenging, opposing the pro-killer activity of the classical activation state [80,81]. Therefore, CDNF suppresses activated microglia by preventing phenotypic polarization to the proinflammatory status, which may also shed light on its underlying mechanisms of improving grafted cell survival and function in 6-OHDA-induced hemiparkinsonian rats. 

Apart from the changes in microglial phenotypes, 6-OHDA stimulation induces ramified microglia shift to amoeboid morphology, which is implicated in neuroinflammatory responses [82,83]. Using a quantitative assessment of BV2 cell outlines, we first observed that the microglia morphology is different after incubation of 6-OHDA for 24 h. Most 6-OHDA-stimulated BV2 cells became amoeboid, accompanied by less ramification in shape. Another striking finding is that co-incubation of 6-OHDA with rhCDNF significantly preserved ramified morphology of Iba1-positive BV2 cells, compared with only 6-OHDA treatment, indicating that CDNF greatly mitigates microglial activation shown in amoeboid shape under pathological condition. Taken together, CDNF treatment was shown to not only regulate microglial polarization, but also its morphological changes in response to 6-OHDA stimulation. Interestingly, RNA sequencing further revealed that CDNF treatment exhibits a strong trend of upregulation of genes related to protein kinase B signaling (Akt signaling pathway), such as Igf1, MerTK, PI3Kcd, and PI3Kcg, in 6-OHDA-stimulated BV2 microglia cells. Murine macrophage and microglia Igf1 production is upregulated by IL-14 or IL13, while downregulated by IFNγ, suggesting that M2-like microgila/macrophage is an important source of Igf1 [84,85]. Moreover, MerTK-dependent activation of PI3K/AKT was shown to inhibit LPS-induced microglial activation and proinflammatory cytokine production [86,87,88]. Accordingly, upregulation of PI3K/AKT pathway by CDNF treatment in 6-OHDA-stimulated BV2 cell, also found in ER-stressed dopaminergic neurons, may facilitate activated microglia polarizing to immunomodulatory phenotype. Next, IPA software indicated that the highly statistically CDNF-regulated canonical pathway is a phagosome formation of BV2 cells in response to stimulation with 6-OHDA, which is activated by identified genes (i.e., PIK3cg, PIK3cd, and ITGA2B). Interestingly, PI3K is a prone signaling pathway to regulate MerTK- or Igf1-mediated phagocytosis of myeloid cells [89,90,91,92]. Taken together, the fact that these changes correlate with microglial phenotype and phagocytic activity may clarify that CDNF treatment improves grafted cell survival and function. 

There were several limitations to the present study. First, although our data indicated that CDNF treatment exhibits an improving effect in allografts cell survival and function in the 6-OHDA rodent model of PD, this model does not reflect the key pathological features or disease progression of human PD. In contrast, the α-synuclein rat model of PD is ideal to characterize DA neuron dysfunction, accompanied by extensive α-syn accumulation and a prominent inflammatory response, which may provide closer investigation of survival, long-term functionality, and integrity of the cells in rVM graft [93]. However, as our previous study showed [94], injection of preformed α-synuclein fibrils to rodent resulted in behavior alternations but did not cause significant decrease in TH-positive fiber density within striatum. Therefore, we have to establish a synucleinopathy PD rodent model that better mimics the disease pathology, including α-synuclein accumulation, inflammation, and progressive DA neuron cell loss, to assess how a pathological host environment affects the transplanted cells, and allow for future studies of CDNF’s regenerative effect. Second, although our data indicated that CDNF treatment exhibits an improving effect in allografts cell survival and function, we need to further determine if CDNF treatment also promotes these newly born DA neurons integrated into existing neurons in the lesioned striatum, which might provide a wider application of the transplantation. Finally, our data demonstrated that CDNF treatment can suppress the microglia/macrophage recruitment and proinflammatory activation, which is implicated in graft-induced inflammatory responses. However, some studies indicated that activated microglia capable of clearing cellular debris and releasing cytokines contributes to the improved nervous tissue integrity [95]. Thus, additional studies are warranted to determine the effects of CDNF administered at different times on microglia/macrophage activation in 6-OHDA lesioning Although CDNF was shown to modulate the phenotype and morphological characteristics in 6-OHDA-stimulated microglia, likely via AKT signaling pathway, further study regarding this regulation is needed to clarify the interaction between CDNF and PI3K. 

In this study, we found that the administration of rhCDNF improves survival and function of VM allografts in a Parkinson rat model. We also confirmed that PET is capable of monitoring neuroinflammation in the grafted striatum by analysis of increasing uptake of [^18^F] FEPPA, which was downregulated by CDNF treatment. The neuroprotective and immunomodulatory effect of CDNF is attributed to limitation of microglia/macrophage proinflammatory activation after transplantation. Moreover, we hypothesize that the modulation of microglial activation by promoting the pro-reparative phenotype polarization and preserving ramified morphology can be explained by the influence over MerTK/PI3K/AKT signaling. IPA analysis among DEGs further provides a novel property of CDNF in augmenting phagosome formation in BV2 cells under stress condition, which enhances our knowledge of the role of CDNF in mediating microglial functions.

## Figures and Tables

**Figure 1 biomedicines-10-01446-f001:**
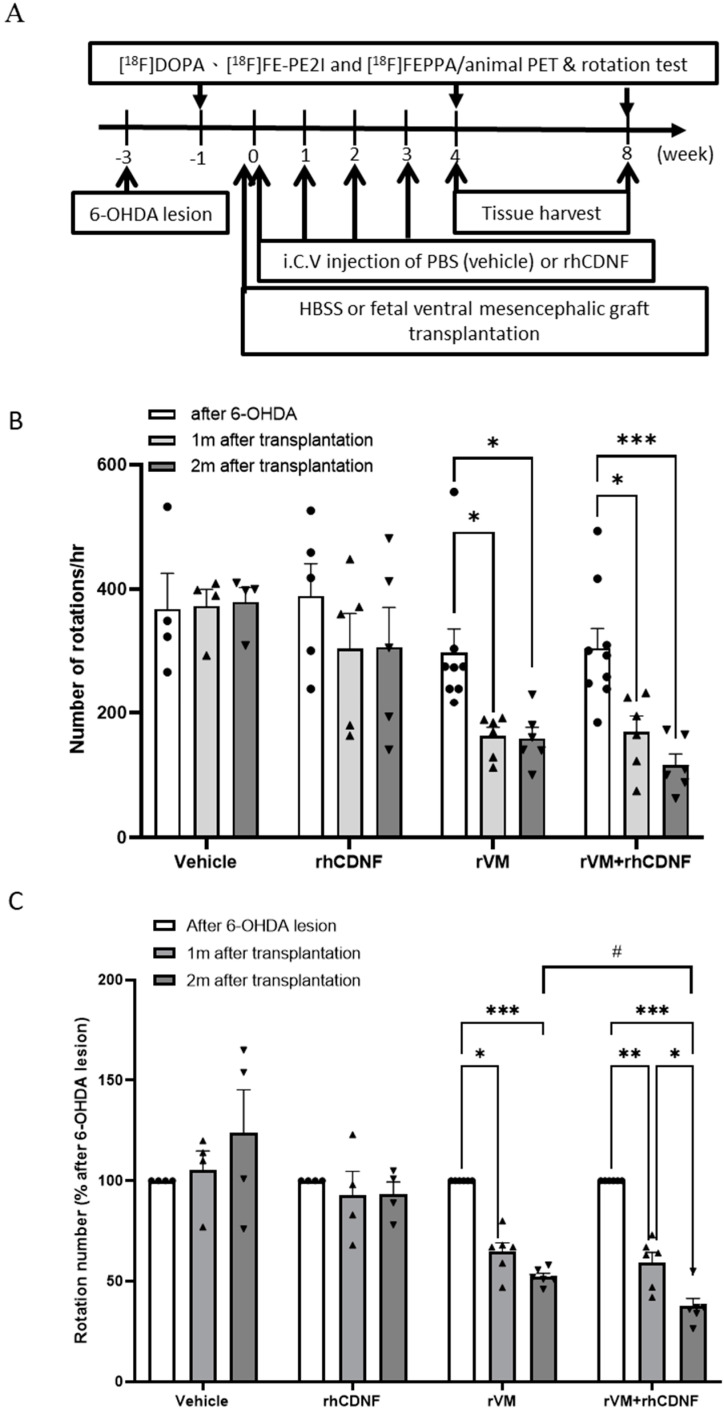
(**A**) Experimental flowchart. [^18^F] DOPA, [^18^F] FE−PE2I, and [^18^F] FEPPA scans were performed at four time points (one before and three after a unilateral 6−hydroxydopamine (6−OHDA) lesion was made to medial forebrain bundle of the animals. Behavior test refers to the apomorphine-induced rotation test. Transplantation was performed three weeks after the 6−OHDA lesion was made. RhCDNF or PBS (vehicle) was administered via intracerebroventricular route at four time points. At 4 and 8 weeks after transplantation, the rats were sacrificed for immunohistochemistry (IHC) studies. (**B**) The apomorphine−induced rotation behavior test was used to evaluate dopaminergic (DA) function of hemiparkinsonian rats that received PBS (vehicle), rhCDNF alone, rVM, or rVM + rhCDNF. The test was performed before and after transplantation. (**C**) Functional recovery of the PD rats with rVM grafting at 4 and 8 weeks after transplantation, as revealed by proportion of apomorphine-induced rotation number after 6−OHDA lesioning to after transplantation. RhCDNF treatment further improved functional recovery of PD rats at 8 weeks after transplantation, compared to only rVM grafting. * *p* < 0.05, ** *p* < 0.01, *** *p* < 0.001 by Tukey’s multiple comparisons test, following two-way ANOVA. # *p* < 0.05 indicates comparison with the rVM + rhCDNF group with two-way ANOVA and Tukey’s post hoc test. The data represent mean ± SEM.

**Figure 2 biomedicines-10-01446-f002:**
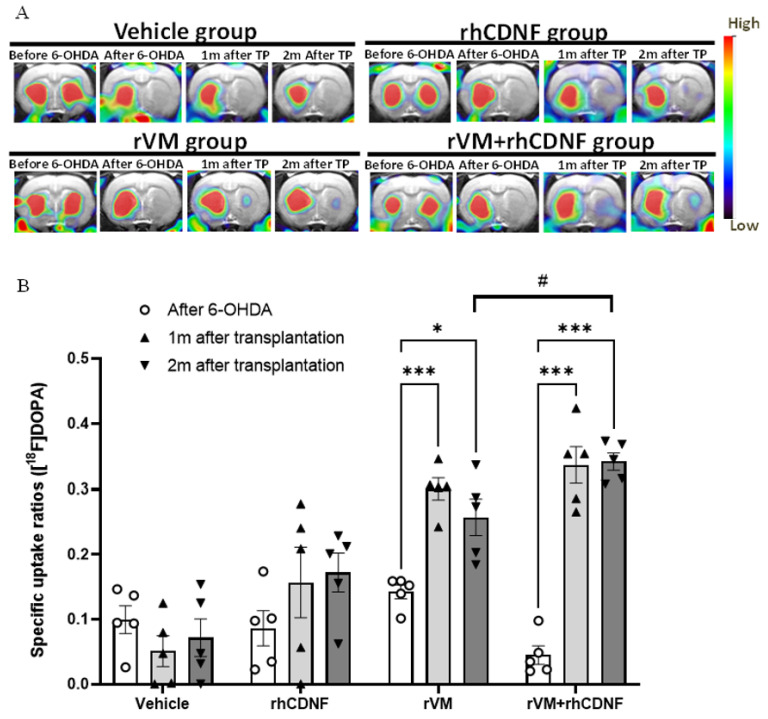
PET images of [^18^F] DOPA uptake distribution in rat brains. (**A**) Coronal (upper panel) and horizontal (lower panel) sections of each group were acquired 50−80 min after injection of ^18^F-DOPA. Left, middle, and right columns of each group represent ^18^F-DOPA uptake before the 6−OHDA lesion, after the 6−OHDA lesion, and at 4 and 8 weeks after transplantation, respectively. (**B**) Specific uptake ratios (SURs) of ^18^F−DOPA of the grafted striatum at different time points and under differ therapeutic regimen. * *p* < 0.05, *** *p* < 0.001 by Tukey’s multiple comparisons test, following two-way ANOVA. # *p* < 0.05 indicates comparison with the rVM + rhCDNF group with two-way ANOVA and Tukey’s post hoc test. The data represent mean ± SEM.

**Figure 3 biomedicines-10-01446-f003:**
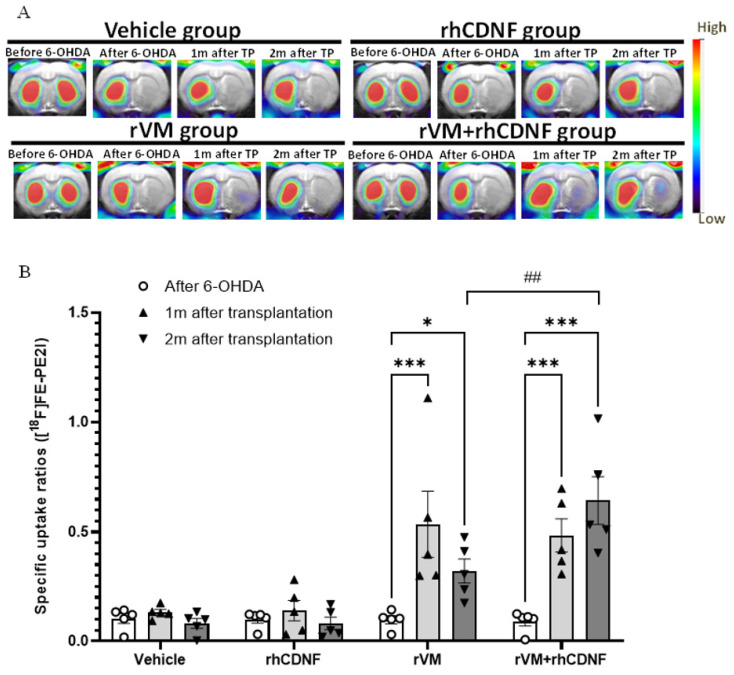
PET images of [^18^F] FE−PE2I uptake distribution in rat brains. (**A**) Coronal (upper panel) and horizontal (lower panel) sections of each group were acquired 20–40 min after injection of ^18^F-FE-PE2I. Left, middle, and right columns of each group represent ^18^F-FE−PE2I uptake before the 6−OHDA lesion, after the 6-OHDA lesion, and at 4 and 8 weeks after transplantation, respectively. (**B**) Specific uptake ratios (SURs) of ^18^F−FE−PE2I of the grafted striatum at different time points and under differ therapeutic regimen. * *p* < 0.05, *** *p* < 0.001 by Tukey’s multiple comparisons test, following two-way ANOVA. ## *p* < 0.01 indicates comparison with the rVM + rhCDNF group with two-way ANOVA and Tukey’s post hoc test. The data represent mean ± SEM.

**Figure 4 biomedicines-10-01446-f004:**
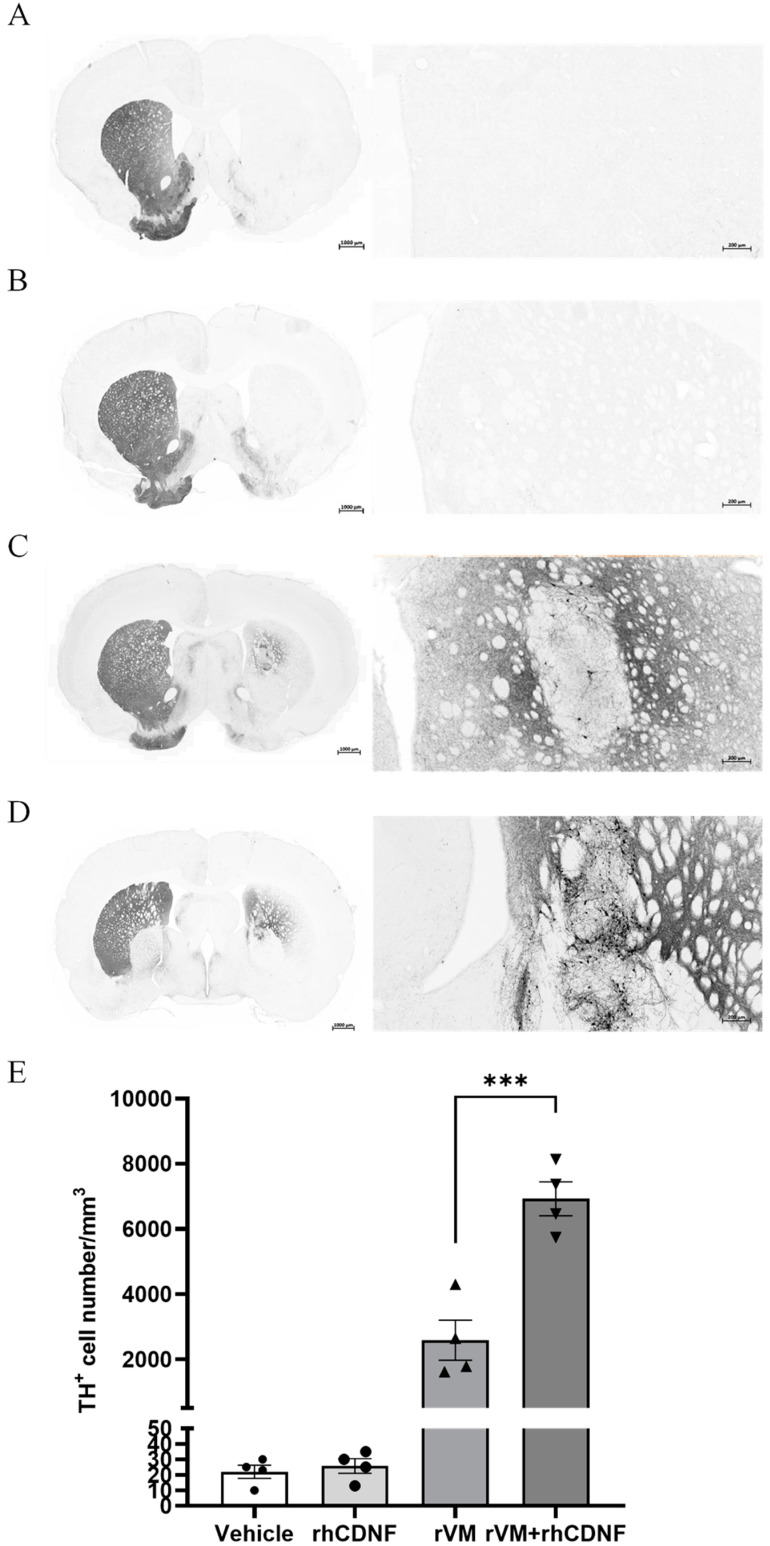
Photomicrographs of tyrosine hydroxylase immunoreactive (TH−ir) cells in hemiparkinsonian rat striatum at 8 weeks following transplantation. TH-ir cell bodies and fibers in the grafted side (right side of coronal brain sections) of the rVM and rVM + rhCDNF groups. Higher density of TH−ir cells was found in the rVM + rhCDNF group as compared to the rVM group. (**A**) Vehicle group. (**B**) rhCDNF alone group. (**C**) rVM group. (**D**) rVM + rhCDNF group. (**E**) Quantification of TH−ir cell density. *** *p* < 0.001 vs. rVM and rVM + rhCDNF. *N*= 4–5 rats in each group.

**Figure 5 biomedicines-10-01446-f005:**
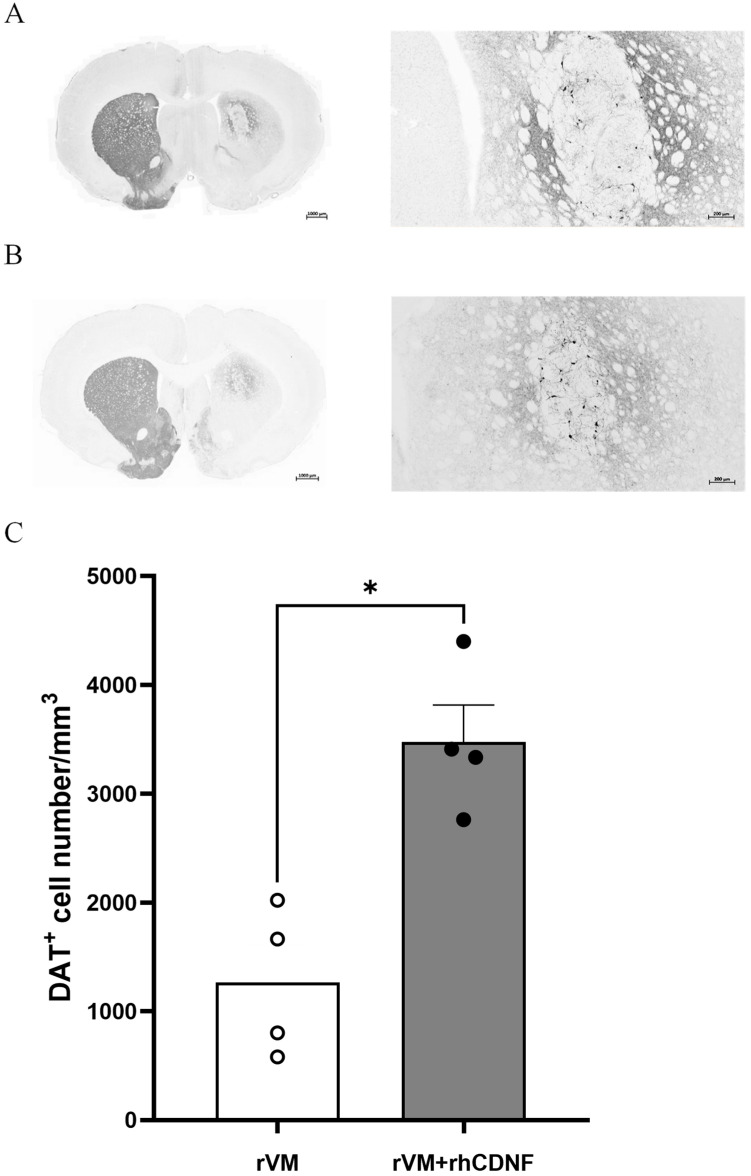
Photomicrographs of DAT immunoreactive (DAT−ir) cells in hemiparkinsonian rat striatum at 8 weeks following transplantation. DAT−ir cell bodies in the grafted side (right side of coronal brain sections) of the rVM and rVM + rhCDNF groups. Higher density of DAT−ir cells was found in the rVM + rhCDNF group as compared to the rVM group. (**A**) Rvm group. (**B**) rVM + rhCDNF group. (**C**) Quantification of DAT-ir cell density. * *p* < 0.05 vs. rVM and rVM + rhCDNF. N = 4 rats in each group.

**Figure 6 biomedicines-10-01446-f006:**
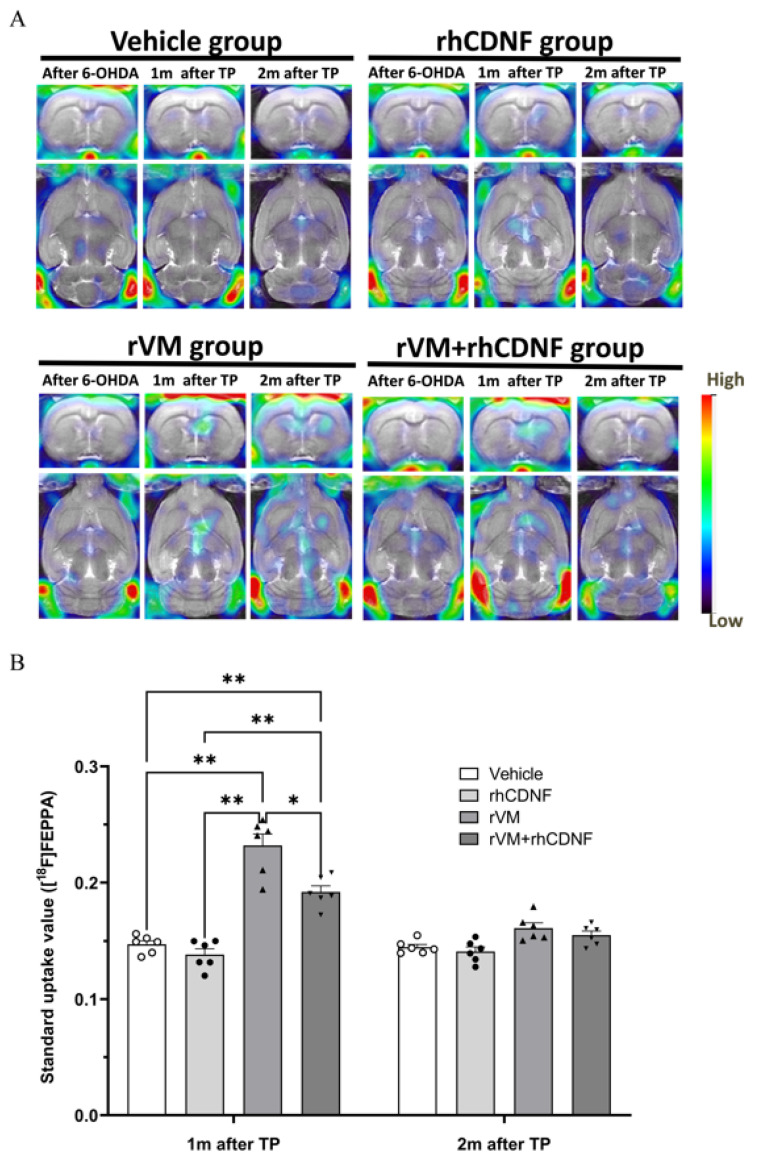
PET images of [^18^F] FEEPA uptake distribution in rat brains. (**A**) Coronal (upper panel) and horizontal (lower panel) sections of each group were acquired after injection of ^18^F−FEEPA. Left, middle, and right columns of each group represent ^18^F-FEEPA uptake after the 6−OHDA lesion, at 4, and 8 weeks after transplantation, respectively. (**B**) Specific uptake ratios (SURs) of ^18^F-FEEPA of the grafted striatum at different time points and under different therapeutic regimen. * *p* < 0.05, ** *p* < 0.01 by Tukey’s multiple comparisons test, following two-way ANOVA. The data represent mean ± SEM.

**Figure 7 biomedicines-10-01446-f007:**
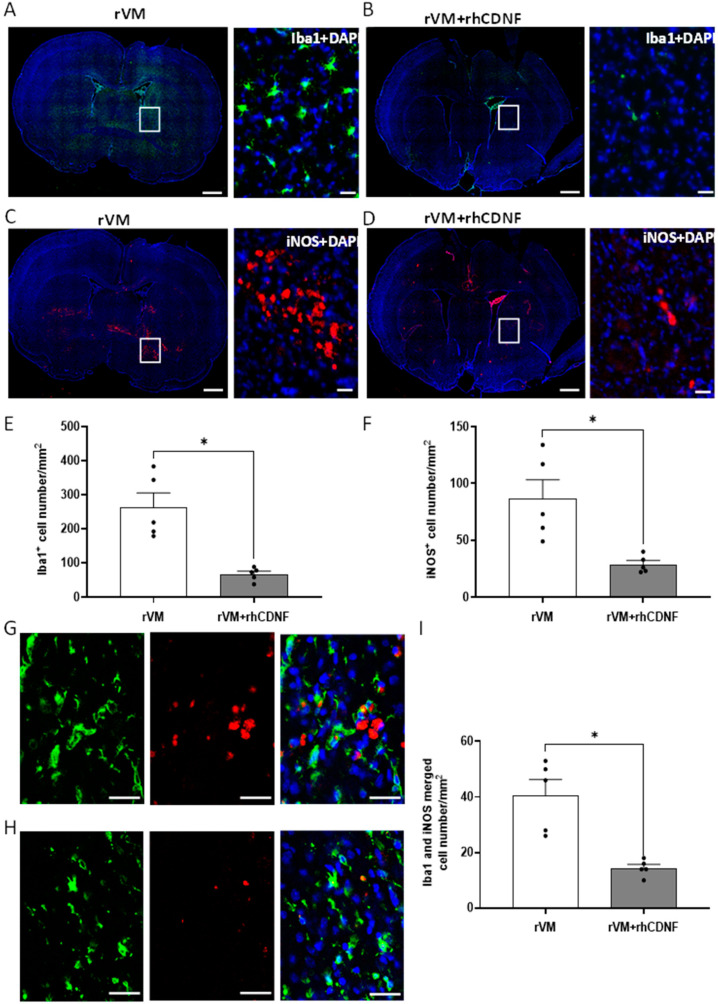
Distribution of microglia/macrophage marker, Iba1, and inflammatory marker in the grafted striatum at four weeks after transplantation. (**A**,**B**) Representative images of grafted striatum sections immunostained with Iba1 and counterstained with DAPI (blue). In the rVM group, there is a large number of Iba1−positive cells in the grafted striatum (**A**), while less Iba1−postive staining was found in the corresponding area of rVM + rhCDNF group (**B**). The right column, representation of panel in left column, shows microglia/macrophage in higher magnification stained with Iba1. Scale bar = 1000 μm in left column of each group. Scale bar = 20 μm in right column of each group. (**C**,**D**) Representative images of grafted striatum immunostained with iNOS. The insets show cells in higher magnification stained with iNOS antibody. Less iNOS-positive staining was found in the grafted striatum of rVM + rhCDNF (**D**) compared to rVM group (**D**). Scale bar = 1000 μm in left column of each group. Scale bar = 20 μm in right column of each group. (**E**) Quantitation of Iba1-positive cells in the grafted striatum at 4 weeks after transplantation showing accumulation of microglia/macrophages in rVM group. In the rVM + rhCDNF group, there is a significant decrease in Iba1−postive microglia/macrophage density in the corresponding area. (**F**) Quantitation of iNOS−positive cells in the grafted striatum at 4 weeks after transplantation showing accumulation of inflammatory cells in rVM group. In the rVM + rhCDNF group, there is a significant decrease in iNOS-positive immune cell density in the corresponding area. (**G**,**H**) Section of grafted striatum at 4 weeks after transplantation double-stained with Iba1 (green) and iNOS (red). rVM + rhCDNF group (**G**) showed fewer Iba1− and iNOS−positive cells compared with the rVM group (**H**). Scale bar = 20 μm. (**I**) Quantification of Ib1−and iNOS−positive cell numbers * *p* < 0.05 indicates comparison with rVM group with Student’s *t*-test. The data represent mean ± SEM.

**Figure 8 biomedicines-10-01446-f008:**
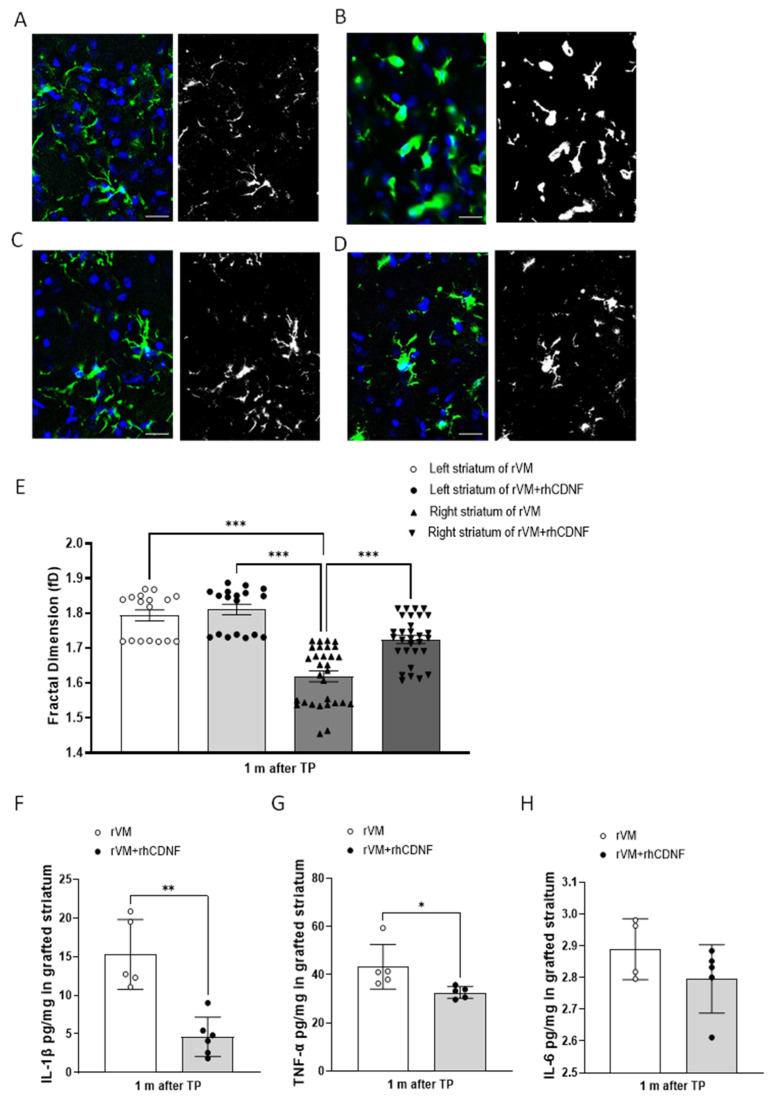
CDNF affects complexity analysis of Iba1−positive cell morphologies and suppresses the production of proinflammatory cytokines in the grafted striatum at one month after transplantation. (**A**–**D**) Fractal analysis of microglia/macrophages in Iba1-stained tissue. Original photomicrographs were subjected to a series of uniform ImageJ plugin protocols prior to conversion to binary images. Binary images were subsequently analyzed by using FracLac of ImageJ, which quantifies single cell complexity (fractal dimension, fD). Representative images of morphological changes in Iba1−staining cells within the left striatum in rVM−grafted rats (**A**), the right striatum in rVM-grafted rats (**B**), the left striatum in rVM + rhCDNF−grafted rats (**C**), and the right striatum in rVM + rhCDNF−grafted rats (**D**). Scale bar = 20 µm. (**E**) Summary data and statistical analysis of fractal dimension at one month after transplantation (*n* = 17−30, one-way ANOVA *p* < 0.001, *** *p* < 0.001 compared to the right striatum in rVM-grafted rats, Tukey’s post hoc test). (**F**) IL-1β concentrations in the grafted striatum were measured by ELISA. (**G**) TNF−α concentrations in the grafted striatum were measured by ELISA. (**H**) IL−6 concentrations in the grafted striatum were measured by ELISA. * *p* < 0.05, ** *p* < 0.01 indicates comparison with rVM group with Student’s *t*-test. The data represent mean ± SEM.

**Figure 9 biomedicines-10-01446-f009:**
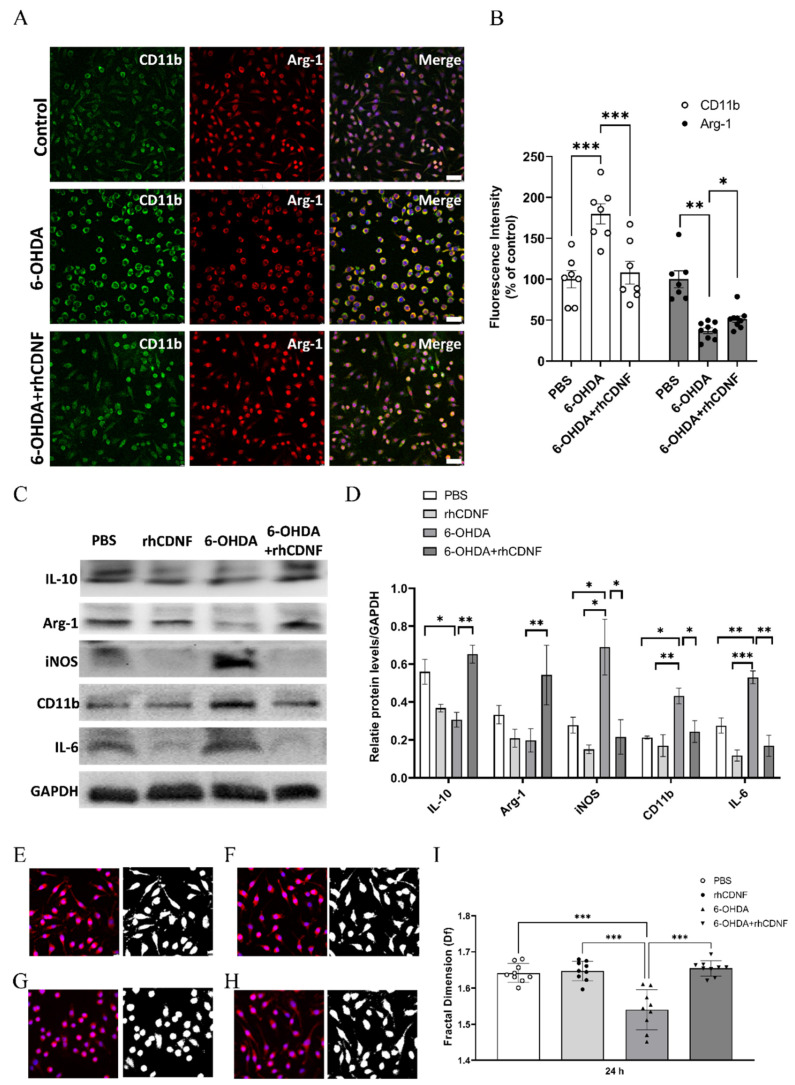
Effects of CDNF on 6−OHDA−stimulated BV2 microglial cells. (**A**) Representative pictures of BV2 cells double-stained for CD11b (green) or Arginase−1 (Arg−1; red), and nuclei were counterstained with DAPI (blue). Photomicrographs were taken with an epifluorescence microscope; scale bar = 50 µm. (**B**) The fluorescent intensities divided by the untreated BV2 cells (control group). There was an increase in CD11b fluorescent intensity in BV2 cells after 6−OHDA treatment as compared with the untreated BV2 cells (control group), and rhCDNF given simultaneously with 6-OHDA decreased activated CD11b intensity in BV2 cells. In contrast, the fluorescent intensity of Arg-1 in 6-OHDA-stimulated BV2 cells was lower than the control group, which was upregulated by CDNF supplementation. (**C**) Lysates from BV2 cells after PBS, rhCDNF, 6-OHDA, or 6-OHDA + rhCDNF treatment were immunoblotted and IL10, Arg−1, iNOS, CD11b, IL−6, and GAPDH levels were analyzed. (**D**) Protein levels were quantified in relation to levels of GAPDH, a housekeeping protein (n = 5, mean ± S.E.M). (**E**–**H**) Morphological analysis of Iba1−stained BV2 cells. The process to prepare photomicrographs for morphological analysis: Original photomicrographs were subjected to a series of uniform ImageJ plugin protocols prior to conversion to binary images. Binary images were then analyzed by using FracLac for ImageJ, which quantifies single cell complexity (fractal dimension, fD). The calculated fD of the cell is shown below its binary image. Representative images of morphological changes in Iba1-staining cells within PBS-treated BV2 cells (**E**), rhCDNF−treated BV2 cells (**F**), 6−OHDA-treated BV2 cells (**G**), and rhCDNF supplementation in 6−OHDA−treated BV2 cells (**H**). (**I**) Summary data and statistical analysis of fractal dimension in 24 h cultured BV2 cells. Fractal dimension was decreased in the 6−OHDA compared with the PBS or rhCDNF alone treatment. However, rhCDNF given simultaneously with 6-OHDA treatment restored the fractal dimension of Iba1−positive cells (*n* = 18−22, one-way ANOVA *p* < 0.001, *** *p* < 0.001, ** *p* < 0.01, * *p* < 0.05 compared to 6-OHDA, Tukey’s post hoc test). The data represent mean ± SEM.

**Figure 10 biomedicines-10-01446-f010:**
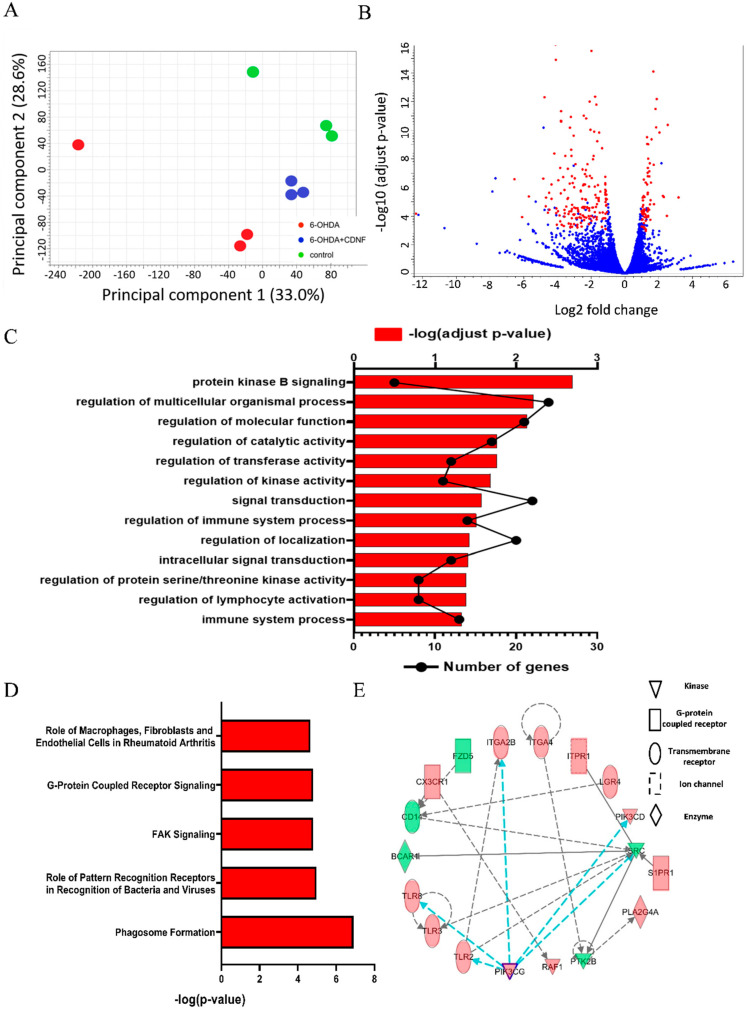
CDNF treatment regulates gene expression in the 6−OHDA−stimulated BV2 microglial cells. (**A**) Principal component analysis (PCA) of the transcriptomes among control, 6−OHDA, and 6−OHDA + rhCDNF groups. (**B**) Volcano plot comparing the log2fold changes and adjusted *p*-values of 21,860 gene expressions. The red dots indicate genes significantly regulated (log2fold change >1 or <−1 adjusted *p*-value < 0.001), and the blue dots indicate genes with no significant change between 6−OHDA and 6−OHDA + rhCDNF. (**C**) GO biological processes overrepresentation analysis based on 253 DEGs. (**D**) Ingenuity^®^ Bioinformatics pathway analysis revealed that highly canonical pathways were differentially expressed in PBS−treated and rhCDNF−treated BV2 cell lines in response to 6−OHDA stimulation. The canonical pathways included in this analysis are shown along the *y*-axis of the bar chart. The *x*−axis indicates the statistical significance. Calculated using the right-tailed Fisher exact test, the *p*−value indicates which biologic annotations are significantly associated with the input molecules relative to all functionally characterized mammalian molecules. (**E**) The activity of highly connected positive regulators of the inflammatory genes PI3KCG, PI3KCD, TLR2, and TLR8 led to the activation of this phagosome formation network, as assessed using the IPA molecule activity predictor in rhCDNF−treated BV2 cells after 24 h 6−OHDA stimulation.

## Data Availability

The datasets supporting the conclusions of this article are included within the article and its Appendix A.

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
