# Peer review of "Modulating Microglia/Macrophage Activation by CDNF Promotes Transplantation of Fetal Ventral Mesencephalic Graft Survival and Function in a Hemiparkinsonian Rat Model"

_biomedicines, 2022, doi:10.3390/biomedicines10061446_

Round 1

Reviewer 1 Report

The authors show that CDNF administration enhances the survival of the grafted dopaminergic neurons and improves functional recovery. Also, this recovery is associated with an increase in dopamine transport expression in the treatment region and a decrease in microglia activation. 

However, these observations are limited due to the PD model used because the 6-OHDA injection is associated with oxidative stress and doesn't show the asynucleopaties features, a key element in therapies for PD based on graft. Many other studies show the possible invasion into the graft of a-synuclein protein in PD patients. Also has been demonstrating the spreading of a-synuclein to other brain regions associated with PD pathogenesis. 

I suggest including a PD animal model based on synuclein overexpression and determining the effect of this kind of graft in recovering the dopaminergic neuronal loss. Another problem with these results is the determination of these new dopaminergic neurons' connectivity with other endogenous neurons and their potential application for a long time.

Is not clear if the CDNF effect is on graft cells, microglia cells, or both in vivo PD models. I suggest including the determination of pro-inflammatory cytokines such as IL-1b, TNFa, IL2, amount others. Also, determine the morphological changes in Iba1 positive cells.

Moreover, how do you explain the modest effect of CDNF on motor behavior compared to the cellular and molecular studies?.

The authors must be including the determinations of dopaminergic neuronal loss in the Sustancia nigra brain region. 

Author Response

Dear Reviewer,

Thank you for your kind letter stating that " the paper is potentially acceptable subject to major revisions that explicitly deal with the comments of the more critical review." Below we detail our response to both reviewers and revisions.

  1. However, these observations are limited due to the PD model used because the 6-OHDA injection is associated with oxidative stress and doesn't show the asynucleopaties features, a key element in therapies for PD based on graft. Many other studies show the possible invasion into the graft of a-synuclein protein in PD patients. Also has been demonstrating the spreading of a-synuclein to other brain regions associated with PD pathogenesis. I suggest including a PD animal model based on synuclein overexpression and determining the effect of this kind of graft in recovering the dopaminergic neuronal loss.                                                  -We thank Reviewer 1 for this important point. The senior author, Dr. Mikko Airavaara, focused on neurodegenerative disease and is prestigious in investigating the pathogenesis and therapeutic strategy of Parkinson's disease in these years. In our latest study published in Molecular Therapy (PMID: 33940158), TH density in the striatum was not affected, while behavior deficit occurred gradually after rats were injected with mouse α-synuclein PFFs. Therefore, we need to establish a severe synucleinopathy PD model, exhibiting α-synuclein accumulation, inflammation, and progressive DA neuron cell loss, to determine the regenerative effects of CDNF in the future study. This explanation is addressed in the discussion (page 32, line 813-824).

  2. Another problem with these results is the determination of these new dopaminergic neurons' connectivity with other endogenous neurons and their potential application for a long time.                                                    - Thanks for the reviewer's comments. Using the current experimental setup, we cannot conclude that new dopaminergic neurons can integrate with other endogenous neurons, while rVM+rhCDNF treatment was shown to alleviate behavior deficit. Additional studies are warranted to determine if CDNF promotes these newly born DA neurons integrated into existing neurons in the lesioned striatum, which might provide a wider application of the transplantation. (page 32, line 825-829)

  3.  Is not clear if the CDNF effect is on graft cells, microglia cells, or both in vivo PD models. I suggest including the determination of pro-inflammatory cytokines such as IL-1b, TNFa, IL2, amount others. Also, determine the morphological changes in Iba1 positive cells.                        -Thanks for the reviewer's comments. We supplement the cytokine experiments in our study (page 18, line 498-505). Also, we provided the morphological changes in vivo study (page 17-18, line 483-497). 

  4. Moreover, how do you explain the modest effect of CDNF on motor behavior compared to the cellular and molecular studies?                            -Thanks for the reviewer's comments. We gave some explantation about the modest effect of CDNF on motor function improvement, as compared with CDNF's regenerative effect on newly born dopamine neuron survival. (page 30, line 699-712)
  5. The authors must be including the determinations of dopaminergic neuronal loss in the Sustancia nigra brain region.                                         -Thanks for the reviewer's comments. We have provided the figures to show the dopaminergic neuronal loss in the Substantia Nigra at 2 months after transplantation (Figure supplement 2).

Reviewer 2 Report

The manuscript entitle “Modulating microglia/macrophage activation by CDNF promotes transplantation of fetal ventral mesencephalic graft survival and function in a hemi-parkinsonian rat model” investigates the protective effect of CDNF in the efficacy of maturation and cell survival of rVM transplantation in PD rat model. Although the paper is well planned, there are some points that should be improved:

Major points:

1.       In my opinion, the main major point that the authors should solved is the nomenclature of microglia polarization because, nowadays M1 and M2 it seems oldated, please see: PMID: 30206328; PMID: 31627485 between others. The authors should be included this information throughout all the manuscript and discuss theirs results in relation with this new microglia nomenclature.

2.       In Fig. 7, the authors show good immunological staining of microglia/macrophages. Since in the cultures they have analyzed the morphology of the BV2 cells by means of Image J (FracLac plugging), it would be very interesting to carry out the same morphology analysis in the striatum of the different groups of rats.

3.       In the results, I have found a contradiction with respect to what is indicated in discussion. I mean, in results the authors indicate the following: "After 6-OHDA injections, the uptake of [18F] FEPPA on the lesion side of striatum was modest, suggesting that 6-OHDA-induced DA denervation in the striatum does not cause considerably inflammatory responses". However, in the discussion they state this: "Apart from the changes in microglial phenotypes, 6-OHDA stimulation induces  ramified microglia shift to amoeboid morphology, which is implicated in neuroinflammatory responses [81, 82]." Could you indicate which of the two statements is the correct and amend it in the text?

Minor points:

1.       Line 175: “The protocol for small animal PET imaging was adapted from previous research.”

Line 214: “The quantification of microglia was performed as previously described. References are required.

2.       In M&M, images quantification (IF and IHQ) should be deeply explained.

3.       In M&M, the precedence of protein should be described.

4.       Next phrase should be rewritten for its better understanding : “Although both rVM and rVM+rhCDNF grafts led to a significant reduction in the ratio of rotation number in the PD model rats, the letter at 2 months after transplantation appeared to produce a profound improvement in rotational asymmetry behavior compare to the former in the rats (Fig. 1C).”

5.       There are some reference with number but without format. For instance: Line383, 693, 704….

6.       The scale bar dimension should be included in the figure legend.

7.       Some word are wrong, for instance: improvs, letter, insect, Df (in spite of fD)…

8.       Many superscripts, such as mm2, are misslocated. For instance: Line412, 476, 479…

Author Response

Dear Reviewer,

Thank you for your kind letter stating that " the the paper is potentially acceptable subject to major revisions that explicitly deal with the comments of the more critical review." Below we detail our response to both reviewers and revisions.

  1. In my opinion, the main major point that the authors should solved is the nomenclature of microglia polarization because, nowadays M1 and M2 it seems oldated, please see: PMID: 30206328; PMID: 31627485 between others. The authors should be included this information throughout all the manuscript and discuss theirs results in relation with this new microglia nomenclature.                                                                                                  - We thank Reviewr 2 for this important point. We totally agree the point that M1/M2 dichotomy nomenclature is old classification for dscribing microglia/macrophage status. Nowaday, many scholars would like to use genomic signature to describe immune cells as DAM or homeostatic microglia. Therefore, we included these informations in introduction (page 4) and adopt pro-inflammatory, disease-associated and pro-reparative microglia to address its phenotype. 
  2.  In Fig. 7, the authors show good immunological staining of microglia/macrophages. Since in the cultures they have analyzed the morphology of the BV2 cells by means of Image J (FracLac plugging), it would be very interesting to carry out the same morphology analysis in the striatum of the different groups of rats.                                                         -We thank Reviewer 2 for this positive comment. We have added morphlogy analysis data of grafted straitum in our manuscript (page 18, figure 8A-E). 
  3.  In the results, I have found a contradiction with respect to what is indicated in discussion. I mean, in results the authors indicate the following: "After 6-OHDA injections, the uptake of [18F] FEPPA on the lesion side of striatum was modest, suggesting that 6-OHDA-induced DA denervation in the striatum does not cause considerably inflammatory responses". However, in the discussion they state this: "Apart from the changes in microglial phenotypes, 6-OHDA stimulation induces  ramified microglia shift to amoeboid morphology, which is implicated in neuroinflammatory responses [81, 82]." Could you indicate which of the two statements is the correct and amend it in the text?                                -Thanks for reviewer's comment! We revised the result ad had more clear interpretation (page 17).
  4. Line 175: “The protocol for small animal PET imaging was adapted from previous research.”  Line 214: “The quantification of microglia was performed as previously described. References are required.                        -Thanks for reviewer's comment! We added the reference in page 8 and page 9.
  5. In M&M, images quantification (IF and IHQ) should be deeply explained.    -Thanks for reviewer's comment! We had more deeply interpretation in page 9-11.
  6. In M&M, the precedence of protein should be described.                              -Thanks for reviewer's comment! We had more deeply interpretation in page 11.
  7. Next phrase should be rewritten for its better understanding : “Although both rVM and rVM+rhCDNF grafts led to a significant reduction in the ratio of rotation number in the PD model rats, the letter at 2 months after transplantation appeared to produce a profound improvement in rotational asymmetry behavior compare to the former in the rats (Fig. 1C).” -Thanks for reviewer's comments! We have repharsed "Although both rVM and rVM+rhCDNF grafts led to a significant reduction in the ratio of rotation number in the PD model rats, rVM+rhCDNF grafts appeared to produce a profound improvement in rotational asymmetry behavior at 2 months after transplantation, compare to the rVM group (Fig. 1C)" in page 14.
  8. There are some reference with number but without format. For instance: Line383, 693, 704…                                                                                          -Thanks for reviewer's comment! We found these reference fomat is correct in Word format. However, when it is transferred to PDF format by editor board, the reference format will be worng!  We will ask the Editor to correct it!
  9.  The scale bar dimension should be included in the figure legend.                -Thanks for reviewer's comment! We have added the bar dimension in the figure 7, 8, and 9 and their legends.
  10. Some word are wrong, for instance: improvs, letter, insect, Df (in spite of fD)…                                                                                                                -Thanks for reviewer's comment! We have revised it in page 26, and figure 9I
  11.  Many superscripts, such as mm2, are misslocated. For instance: Line412, 476, 479…                                                                                                       -Thanks for reviewer's comment!We found these superscripts are correct in Word format. However, when a manuscript is transferred to PDF format by editor board, the superscripts will be mislocated!  We will ask the Editor to correct it! 

Round 2

Reviewer 1 Report

The authors provide new antecedents to improve the manuscript. I suggest exploring the efficacy of this therapy in a-synuclein models in the future studies

Reviewer 2 Report

All requested revisions have been carried out by the authors.